# Loss of ANCO1 Expression Regulates Chromatin Accessibility and Drives Progression of Early-Stage Triple-Negative Breast Cancer

**DOI:** 10.3390/ijms241411505

**Published:** 2023-07-15

**Authors:** Meng Yuan, Megan E. Barefoot, Kendell Peterson, Moray J. Campbell, Jan K. Blancato, Manjing Chen, Marcel O. Schmidt, Amber J. Kiliti, Hong-Bin Fang, Anton Wellstein, Anna T. Riegel, Ghada M. Sharif

**Affiliations:** 1Lombardi Comprehensive Cancer Center, Georgetown University, Washington, DC 20057, USA; 2Department of Biomedical Sciences, Cedars-Sinai Medical Center, Los Angeles, CA 90048, USA; 3Department of Biostatistics, Bioinformatics and Biomathematics, Georgetown University, Washington, DC 20057, USA

**Keywords:** ANCO1, ANKRD11, TNBC, tumor suppressor, breast cancer enhancer

## Abstract

Mutations in the gene ankyrin repeat domain containing 11 (*ANKRD11*/*ANCO1*) play a role in neurodegenerative disorders, and its loss of heterozygosity and low expression are seen in some cancers. Here, we show that low ANCO1 mRNA and protein expression levels are prognostic markers for poor clinical outcomes in breast cancer and that loss of nuclear ANCO1 protein expression predicts lower overall survival of patients with triple-negative breast cancer (TNBC). Knockdown of ANCO1 in early-stage TNBC cells led to aneuploidy, cellular senescence, and enhanced invasion in a 3D matrix. The presence of a subpopulation of ANCO1-depleted cells enabled invasion of the overall cell population in vitro and they converted more rapidly to invasive lesions in a xenograft mouse model. In ANCO1-depleted cells, ChIP-seq analysis showed a global increase in H3K27Ac signals that were enriched for AP-1, TEAD, STAT3, and NFκB motifs. ANCO1-regulated H3K27Ac peaks had a significantly higher overlap with known breast cancer enhancers compared to ANCO1-independent ones. H3K27Ac engagement was associated with transcriptional activation of genes in the PI3K-AKT, epithelial–mesenchymal transition (EMT), and senescence pathways. In conclusion, ANCO1 has hallmarks of a tumor suppressor whose loss of expression activates breast-cancer-specific enhancers and oncogenic pathways that can accelerate the early-stage progression of breast cancer.

## 1. Introduction

The triple-negative breast cancer (TNBC) subtype is diagnosed in 10–15% of breast cancer patients [1]. It is one of the most aggressive molecular subtypes of breast cancer with a low survival rate and poor outcomes due to the lack of effective targeted therapies. Key drivers and prognostic biomarkers for TNBC tumor progression are still largely lacking.

One possible candidate gene involved in early-stage breast cancer progression is the ankyrin repeat domain containing 11 (*ANKRD11*/*ANCO1*). Inherited mutations in *ANCO1* have been associated with the neuronal development disorders KGB and Cornelia de Lange syndrome [2,3]. ANCO1′s role as a putative tumor suppressor was initially based on the observed loss of heterozygosity at its genomic location, chr16q24.3 [4,5]. Expression of ANCO1 is significantly lower in breast cancer than in normal breast tissue [6]. Loss of ANCO1 expression in breast cancer has been attributed to frequent genomic deletions as well as methylation on its promoter region that contains abundant CpG islands [7,8]. Recent Sleeping Beauty transposon screens revealed ANCO1 as a top hit and a potential tumor suppressor in several models of oncogene-induced mammary cancer [9].

ANCO1 is a large ~300 kDa protein that binds and regulates the activity of a number of transcriptional regulators [10]. It interacts with histone acetylation modifying proteins, such as histone deacetylase 3 (HDAC3) [10,11] and the p300/CBP-associated factor acetyltransferase complex [12]. ANCO1 also acts as a transcriptional repressor that can bind to the p160 nuclear receptor coactivators, including amplified in breast cancer 1 (AIB1/SRC3/NCOA3). ANCO1 interacts with the N-terminal basic helix–loop–helix Per-Arnt-Sim (bHLH-PAS) domain of AIB1 and inhibits transcription by recruiting HDACs [11,13]. We and others have shown that ANCO1 acts as a transcriptional repressor when it binds to AIB1 and is able to suppress hormone-dependent transcription, including that of the HER2 gene [13]. Other studies have shown that ANCO1 can enhance the TP53 tumor-suppressive function [14]. ANCO1 also interacts with AIB1 when it is part of the oncogenic TEAD/YAP complex and can repress TEAD/YAP-dependent expression of a number of genes, including those located on the chromosome 1q21.3 cytoband, which is amplified in breast cancer and has been associated with cancer recurrence and poor clinical outcomes [6,15]. 

Here, we present the first analysis of how ANCO1 protein levels in different subtypes of breast cancer relate to clinical outcomes. Furthermore, we examine the functional effects of loss of ANCO1 expression in vitro and in vivo as well as on gene expression and breast cancer enhancer activation. We found that high levels of nuclear ANCO1 predicted more favorable outcomes, especially in the TNBC subtype, and that loss of ANCO1 expression drove early-stage TNBC cells to more malignant phenotypes both in vitro and in vivo. Loss of ANCO1 expression increased breast cancer gene enhancer activation and, in parallel, activated gene expression patterns related to oncogenic pathways. Taken together, our data indicate that ANCO1 has an important tumor-suppressive function and acts as a chromatin remodeler in early-stage TNBC.

## 2. Results

### 2.1. High ANCO1 Expression Is a Positive Prognostic Indicator in Breast Cancer Patients

We have previously shown that the loss of ANCO1 transcriptional repression at the AIB1-YAP co-activation complex mediates malignant progression of early-stage breast cancer [6]. Here, we examine the association of ANCO1 expression with survival outcomes in patients with different breast cancer subtypes and evaluate the tumor-suppressive mechanism of ANCO1 in TNBC. 

We first examined publicly available datasets (kmplot.com) of patients with different subtypes of breast cancer (ER+/−, PR+/−, and HER2+/−), regardless of age at diagnosis or treatment status, for ANCO1 mRNA expression levels. We found that a high level of ANCO1 was correlated with a more favorable prognosis among all breast cancer subtypes (Figure 1A). Interestingly, overall levels of ANCO1 mRNA expression in different subtypes of breast cancer in the TCGA and Metabric datasets [16,17] showed significantly higher ANCO1 levels in TNBC (ER-, PR-, and HER2-) patient samples compared to other subtypes (Appendix A). Quite strikingly, an overall survival analysis of a cohort of TNBC patients without chemotherapy treatment showed a significant correlation between high ANCO1 mRNA and longer overall survival (Figure 1B), suggesting that ANCO1 expression could be more functionally relevant in TNBC tumors.

We next examined if ANCO1 protein levels also correlated with patient outcomes. For this, we analyzed a breast cancer tissue microarray (TMA) that comprised three immunohistochemistry (IHC) subtypes: ER-positive (ER+, PR+/−, and HER2−), HER2-positive (HER2+, ER+/−, and PR+/−), and TNBC. Each subtype had 100 cores derived from 50 patients. The metadata for this TMA [18] are summarized in Appendix A. The ANCO1 antibody was optimized for IHC on xenograft tumor tissues from control and ANCO1 knockdown cells (Appendix A). The TMA-stained cores were scored for ANCO1 expression levels using the inForm software to determine high versus low expression of ANCO1 (Figure 1C). Both nuclear and cytoplasmic ANCO1 protein levels, summarized as interquartile ranges (IQRs), correlated with better overall and/or recurrence-free survival in all subtypes of breast cancer combined (Figure 1D and Appendix A). Next, we investigated ANCO1 protein levels in each breast cancer subtype and found a significant correlation between high nuclear ANCO1 levels and better overall survival only in the TNBC patient cohort (Figure 1E and Appendix A). Since nuclear localization of ANCO1 is essential for its role as a repressor of oncogenic transcriptional programs, we hypothesized that this observation was likely related to its role as a tumor suppressor [6,11,13].

### 2.2. Reduction in ANCO1 Causes Abnormal Cell Morphology and Leads to Aneuploidy and Senescence

To understand the role of ANCO1 in the initiation and progression of breast cancer, we used the human immortalized mammary epithelial cell line MCF10A (10A) as a model of the pre-malignant stage [19] and the MCFDCIS (DCIS) cell line as a model for early-stage triple-negative ductal carcinoma in situ [20]. Both cell lines are ER- and PR-negative, with no HER2 amplification. We generated ANCO1 knockdown in 10A and DCIS cell lines by lentiviral infection of shANCO1 constructs targeting different regions of ANCO1 mRNA. A decrease in ANCO1 mRNA expression in shANCO1 versus shCTRL cells was confirmed with RT-qPCR (Figure 2A). ANCO1 protein levels were also decreased in shANCO1 cell lines (Figure 2B,C). Since the ANCO1 protein level in DCIS cells was below the detection threshold on Western blots, immunofluorescence (IF) staining was used instead to verify reductions in ANCO1 protein levels in the DCIS shANCO1 cell lines (Appendix A).

In two-dimensional (2D) tissue cultures, the control 10A and DCIS cells were round-shaped and homogenous in size, whereas the shANCO1 cells were heterogenous with enlarged and elongated cells. Cell extensions were also observed in shANCO1 cell lines, suggesting a more invasive phenotype. A dramatic morphological change was seen in DCIS compared to 10A shANCO1 cell lines (Figure 2D and Appendix A). Additionally, both 10A and DCIS shANCO1 cell lines showed increased overall cell size and many of the largest shANCO1 DCIS cells had more than one nucleus (Figure 2D,E). This could be due to the cells’ inability to complete cell division. To test for aneuploidy, we performed fluorescence in situ hybridization (FISH) analysis in two shANCO1 DCIS cell lines using chromosome 17 alpha satellite probe (XCE17) (Figure 2F) [21]. Each signal represents one copy of chromosome 17. ANCO1 knockdown cell lines had a significantly higher percentage of 4N cells than control cells (Figure 2G, Appendix A). As increased size and multinucleation are characteristics of senescent cells [22], we examined ANCO1 knockdown cell lines for senescence-associated (SA) β-galactosidase [23]. Significantly higher levels of positive β-galactosidase staining were observed in shANCO1 10A and DCIS compared to control cells, especially in enlarged cells (Figure 2H,I). In addition to the induction of senescence, ANCO1 knockdown also caused a slight increase in apoptosis in one of the shANCO1 cell lines (Appendix A). 

### 2.3. Reduction in ANCO1 Expression Leads to Aberrant Sphere Formation and Enables Invasion In Vitro

To examine the contribution of ANCO1 knockdown to the malignancy of epithelial cells, we tested the ability of shANCO1 10A and DCIS cells to form mammary acini in three-dimensional (3D) basement membrane cultures [24,25] (Appendix A). Loss of epithelial cell polarity and filling of the luminal space of mammary ducts are known hallmarks of early-stage breast cancer progression [26]. Control 10A cells formed homogenous round spheres with defined borders, yet upon ANCO1 knockdown, large and irregular spheres were observed (Figure 3A). Similarly, ANCO1 knockdown in DCIS cells resulted in aberrant spheres that lost circularity compared to control DCIS spheres (Figure 3A). An increase in sphere size was observed in both 10A and DCIS lines (Figure 3B). To examine the invasive phenotype further, we aggregated DCIS cells in U-shaped wells before they were embedded in a collagen I–Matrigel mix to monitor their ability to protrude and leave the spheres (Appendix A). ANCO1 knockdown in DCIS cells resulted in an enhanced invasion of cells out of the spheres and into the matrix (Figure 3C,D). 

### 2.4. Loss of ANCO1 Expression Enables Collective Invasion in 3D Spheres

We and others have previously shown that ANCO1 can bind and repress the oncogenic transcriptional coactivator AIB1 [6,11,13]. A more oncogenic isoform of AIB1, termed AIB1Δ4, lacks the N-terminal domain of full-length AIB1 and no longer binds ANCO1 [13]. Consequently, AIB1Δ4 does not interact with ANCO1 and functions as a more potent transcriptional coactivator than full-length AlB1, since it escapes ANCO1 transcriptional repression [13,27]. Cells expressing AIB1Δ4 show enhanced invasion and, even as a subpopulation, can enable surrounding cancer cells to collectively invade and metastasize to the lungs [28]. We postulated that loss of ANCO1 repression of the full-length AIB1 would cause AIB1 de-repression and consequently mimic some aspects of the enabler phenotype in 3D. To investigate whether ANCO1 knockdown cells can crosstalk with surrounding cells to enable their invasion, we mixed shANCO1 DCIS cells that were labeled with a red fluorescent protein with unlabeled parental DCIS cells at a 1:4 ratio. The mixed cells were aggregated and embedded in a collagen I-Matrigel mix. ANCO1 knockdown DCIS cells showed an invasive phenotype, and, consistent with our previous observations of the enabling phenotype, a number of parental cells followed the invasive shANCO1 cells out of the spheres and into the matrix (Figure 3E). Parental cells in the mixed culture with shANCO1 cells traveled longer distances from the spheres than those that were cultured alone (Figure 3E,F). These observations indicate that ANCO1 loss in a small proportion of cells may have a field effect on surrounding cell populations that results in enabling a collective invasion. 

### 2.5. Loss of ANCO1 Expression Enhances Invasion In Vivo

Next, we examined the effect of ANCO1 reduction on tumor growth and progression in vivo. We used DCIS cells which give rise to luminal and myoepithelial cell populations in xenografts and progress from ductal carcinoma in situ to invasive ductal carcinoma within a predictable time course [29,30]. The DCIS cells were injected into the flanks of athymic nude mice, and tumors were collected and analyzed 48 days after injection (Figure 3G). As expected, shANCO1 tumors had less ANCO1 protein expression overall (see Appendix A). Histopathological analysis of tumors arising from control or shANCO1 DCIS cells showed an increased percentage of invasive lesions in ANCO1-low tumors (Figure 3H,I). These in vivo results are consistent with our in vitro findings that ANCO1 reduction enhances invasion and therefore contributes to the invasive progression of breast cancer.

### 2.6. ANCO1 Reduction Increases Chromatin Accessibility and Promotes H3K27Ac Binding at Known Breast Cancer Enhancer Regions 

To uncover the epigenetic mechanisms underlying ANCO1-regulated phenotypes, we examined how ANCO1 reduction affected global H3K27Ac distribution on the chromatin. H3K27Ac is a histone modification that marks open and transcriptionally active chromatin regions [31]. We performed ChIP-seq using an antibody targeting H3K27Ac in shANCO1 cell lines and their control counterparts. Differential binding events were established by comparing the degree of H3K27Ac chromatin engagement between shANCO1 and control groups (Appendix A). This analysis identified 15,346 and 4053 H3K27Ac peaks that were significantly up-regulated in shANCO1 10A and DCIS cells, respectively. In comparison, only 43 and 1090 peaks were found to be significantly down-regulated in shANCO1 10A and DCIS cells (Figure 4A). These ANCO1-regulated peaks (see Appendix A) suggest that ANCO1 plays a role in regulating chromatin architecture, leading to a net overall increase in chromatin accessibility in both cell lines that represent different stages of disease progression. We next mapped the distribution of read counts composing ANCO1-regulated peaks with respect to the center of established breast cancer enhancer peaks [32]. ANCO1-regulated peaks had a significantly increased overlap with breast cancer enhancer peaks compared to ANCO1-independent ones which were present in the shCTRL group and did not change in response to ANCO1 knockdown (*p* < 0.0001 for both 10A and DCIS lines, chi-squared test) (Figure 4B). The presence of down-regulated peaks in shANCO1 DCIS cells was likely because the DCIS line, which already progressed to early-stage TNBC, had a decreased ANCO1 level at baseline before knockdown compared to the 10A line.

We next compared ANCO1-regulated H3K27Ac peaks between 10A and DCIS cell lines and found 2792 peaks that overlapped by at least 1bp (Figure 4C). Genes annotated to these ANCO1-regulated peaks were then analyzed by the Database for Annotation, Visualization and Integrated Discovery (DAVID) for signaling pathway enrichment (Appendix A). Consistent with our previous finding that ANCO1 regulates TEAD/YAP-dependent transcription [6], the HIPPO signaling was enriched in ANCO1-regulated peaks in both 10A and DCIS cell lines. Several other pathways, including the cellular senescence pathway, the focal adhesion pathway, and PI3K-AKT signaling, were also commonly enriched (Figure 4D), which matched with the senescent, enlarged, and invasive cell phenotype after ANCO1 depletion [33,34]. We then used HOMER to assess motif enrichment in the differentially bound H3K27Ac peaks that were up-regulated in the ANCO1-depleted 10A and DCIS cells. The top enriched motifs include those of the AP-1 family of transcription factors that mediate many cellular processes in cancer progression, including cell survival, proliferation, and invasion [35,36]. Also, among the significantly enriched motifs are the TEAD family (Figure 4E), which are effector transcription factors of the HIPPO/YAP signaling pathway [37]. TEAD target genes are associated with tumor growth, disease progression, and therapy resistance in breast cancer [38,39]. Global co-occupancy of TEAD/YAP and AP-1, which cooperatively drive tumorigenesis, has been reported [40]. STAT3 and NFκB motifs were also significantly enriched (Figure 4E). These factors are both known to be involved in the inflammatory response that enhances cancer progression [41,42]. (See Appendix A for complete lists of discovered motifs).

### 2.7. Loss of ANCO1 Expression Leads to Activation of Breast Cancer Progression Genes 

We next examined how H3K27Ac enhancement driven by ANCO1 knockdown affected gene expression. We performed an RNA-seq analysis of shANCO1 and shCTRL 10A and DCIS cells in 2D cultures. Comparable patterns of gene expression changes were observed among three different shRNAs for ANCO1 in triplicates (Appendix A). The inflammation regulatory genes *S100As*, *IL1β*, and *SERPINB2*, as well as the keratinization-associated genes *SPRRs* and *IVL*, were up-regulated in 10A and DCIS cells as a consequence of ANCO1 knockdown. The oncogenic senescence-related gene *F3* was also significantly increased in shANCO1 DCIS cells (Appendix A, Appendix A) [43]. Gene Set Enrichment Analysis (GSEA) indicated that the inflammatory response, senescence, and chr1q21.3 gene sets were significantly enriched in ANCO1 knockdown 10A and DCIS cells (Appendix A). The mTORC1 signaling was enriched in shANCO1 10A cells, which could be responsible for the observed cell size increase (see Figure 2D,E) [33]. The epithelial–mesenchymal transition (EMT) gene set was enriched in DCIS cells upon ANCO1 knockdown, which could explain the invasive phenotype we observed in shANCO1 DCIS spheres (see Figure 3C,D). (See Appendix A for complete lists of regulated pathways).

To confirm the ANCO1 knockdown effect on gene expression and the invasive phenotype, we also performed RNA-seq on 3D spheres that were grown in Matrigel for 5 days (Appendix A, design in Appendix A). Genes associated with cell invasion, EMT, and senescence, including *ME1*, *SCEL*, and *TOP2A*, were differentially expressed in shANCO1 cells (Appendix A, Appendix A) [44,45,46,47]. Consistent with signaling pathway changes observed in shANCO1 cells cultured in 2D, the mTORC1 pathway was enriched in ANCO1 knockdown spheres, which was driven by the up-regulation of PLK1 in DCIS spheres (Appendix A, Appendix A). 

### 2.8. ANCO1 Reduction Promotes Transcriptional Activation through Chromatin Remodeling

To determine whether the up-regulation of gene expression was directly attributable to H3K27Ac engagement at corresponding regulatory DNA elements, we annotated the differential H3K27Ac peaks to the nearest transcription start site (TSS) and compared the corresponding genes to differentially expressed genes discovered in RNA-seq. We observed 1751 (10A) and 611 (DCIS) significantly up-regulated genes with enriched H3K27Ac ChIP peaks (Figure 5A), and over 90% of these peaks overlapped with known breast cancer enhancer sites by at least 1bp. The rest of the H3K27Ac ChIP peaks that did not overlap with known breast cancer enhancer sites may be de novo regulatory elements that play a role in driving the malignant progression of breast cancer. The identified up-regulated genes with H3K27Ac engagement included the inflammatory response genes *IL1α*, *IL1β*, and *SERPINB2*; the *HBEGF* gene that mediates cell migration and invasion; *PLAU*, which is regulated by AP-1 and drives cancer metastasis; and *LAMC2*, which is involved in the EMT process [48,49,50]. The senescence gene *F3* and *S100A2* and *IVL*, located on the chromosome 1q21.3 cytoband, were also identified as having corresponding H3K27Ac engagement (Figure 5A). Typical concurrent enrichment of RNA-seq reads with H3K27Ac peaks at corresponding regulatory DNA elements is depicted on selected genes (Figure 5B). The close concurrence of H3K27Ac engagement and gene activation suggests a direct transcriptional regulatory effect of ANCO1 loss on these genes through chromatin remodeling.

Ingenuity Pathway Analysis (IPA) of these identified directly up-regulated genes (log_2_FC > 1.2) with enriched H3K27Ac engagement revealed the S100 family, STAT3, and tumor microenvironment pathways to be enriched in both 10A and DCIS ANCO1 knockdown cells (Appendix A), indicating a direct impact of ANCO1 loss on these pathways in different stages of breast cancer progression. Additionally, the senescence and inflammasome pathways were found to be enriched in shANCO1 DCIS cells (Figure 5C). And, consistent with the discovery in annotated ANCO1-regulated H3K27Ac peaks (see Figure 4D), the PI3K-AKT signaling was up-regulated in shANCO1 DCIS cells and could be responsible for the increased cell size we observed (see Figure 2D,E) [33].

## 3. Discussion

In this study, we analyzed for the first time both mRNA and protein ANCO1 levels in tumor specimens from breast cancer patients covering all three major subtypes. Our data confirm the significance of ANCO1 protein and mRNA levels as prognostic markers and further emphasize their role in the TNBC subtype. We show that nuclear ANCO1 is a more robust predictor of patient outcomes than cytoplasmic ANCO1, suggesting that at least part of ANCO1′s role as a tumor suppressor relies on its interaction with transcriptional regulatory complexes in the nucleus. Indeed, we and others have shown that ANCO1 can alter transcription by interacting with histone acetylation-modifying proteins [10,11,13] and binding and suppressing oncogenic transcriptional complexes, such as TEAD/YAP/AIB1, to halt tumor initiation and progression [6]. 

We now also demonstrate that the loss of ANCO1 has functional effects on oncogenic progression. We utilized the human non-cancerous mammary epithelial cell line 10A to represent the pre-malignant stage [19]. The DCIS cell line was derived from xenograft tumors formed by oncogene-transfected 10A cells and was used in this study as a model for early-stage ductal carcinoma in situ [20]. Both 10A and DCIS are triple-negative cell lines characterized by ER-negative and PR-negative status without HER2 amplification. Reduction in ANCO1 in 10A and DCIS cells induces aspects of the malignant phenotype, such as senescence and aneuploidy, that are known to contribute to cancer progression [51]. Aneuploidy is an established mechanism by which normal cells experience genomic instability and continue to gain mutations that lead to metastasis and therapy resistance [52,53]. Our results showing ANCO1 reduction leading to aneuploidy are consistent with previous reports that demonstrate ANCO1 as a chromatin regulator in neural development and autism [10]. Additionally, ANCO1 has been found to be in complex with cohesin, whose maintenance is essential for chromosome segregation during mitosis [54,55]. ANCO1 localizes to chromatin during mitosis and is degraded by the end of the process [56]. Therefore, disruption of ANCO1 in the cohesin complex could result in genome instability and aneuploidy. Our observations agree with previous findings that show that ANCO1 suppresses multinucleation driven by mutant TP53 [57]. 

While cellular senescence and cell cycle arrest are generally considered protective mechanisms for cells during DNA repair, a senescence-associated secretome has been shown to enhance pro-tumor inflammation [58]. Consistently with ANCO1-low phenotypes, mTOR inhibition by rapamycin abrogated the pro-inflammatory secretome by senescent cells [59]. As TNBC cells lose ANCO1 expression during cancer progression, their crosstalk with surrounding cell populations may facilitate a field effect that can enhance malignant programs. This is one possible mechanism by which ANCO1-low cells drive the bulk population to invade in mixed 3D spheres. This hypothesis warrants further investigation, since determining the factors and pathways responsible for this enabling crosstalk could help design targeted therapies that inhibit the tumor-initiating effects of cells that have lost ANCO1 expression.

Abnormal enhancer activities are broadly detected in cancers, where they drive enhanced oncogene activation, leading to tumorigenesis. This oncogenic enhancer pattern is tumor-type-specific and accompanied by aberrant activities of transcriptional regulator proteins [60]. We show that loss of ANCO1 results in globally activated breast cancer enhancers, suggesting that ANCO1 may serve as a master suppressor for oncogenic transcriptional programs in breast cancer and that its loss potentially contributes to disease progression by controlling a variety of signaling mechanisms. It would be interesting to further characterize these ANCO1-regulated enhancers and associate them with the progressive phenotype of the ANCO1-low cell subpopulation. In addition, we observed a smaller number of ANCO1-regulated H3K27Ac peaks in the DCIS line compared to the 10A line as well as the presence of down-regulated peaks in the shANCO1 DCIS line. DCIS cells are malignant early-stage breast cancer cells with a low expression of ANCO1, and therefore ANCO1-mediated chromatin remodeling might have already impacted the H3K27Ac distribution in DCIS cells at the baseline. 

Here, we show that the up-regulated gene expression in oncogenic signaling is accompanied by elevated H3K27Ac signals at corresponding gene loci. This may indicate that ANCO1′s role as a transcriptional co-repressor involves inducing chromatin remodeling. Recently published Sleeping Beauty screens aiming at identifying genes involved in breast cancer initiation and progression characterized *ANCO1* as a frequently lost gene in genetically engineered mouse models of Pik3caH1047R-, KrasG12D-, and Stat3c-induced breast cancer, emphasizing its suppressive role that hinders oncogene-driven aberrant transcription [9]. It is interesting, then, that the PI3K-AKT signaling pathway is implicated in up-regulated gene regions with increased H3K27Ac signaling due to ANCO1 loss and mTOR is one of the consistently enriched pathways in low ANCO1 cells in 2D and 3D models. This observation suggests a synergistic effect of ANCO1 loss and oncogenic PI3K-mTOR signaling activation in driving the early progression of breast cancer. In fact, the EMT regulatory factor LAMC2, which is activated upon ANCO1 knockdown, has been reported to promote the expression and phosphorylation of PI3K [50]. There is also evidence that YAP can mediate the crosstalk between Hippo signaling and PI3K-mTOR signaling [61]. In the current study, we have further highlighted a suppressive role of ANCO1 in this oncogenic signaling network.

Overall, we have confirmed a high ANCO1 level as a positive indicator for clinical outcomes in breast cancer patients and its loss as a potential driver of TNBC progression. We have further revealed possible epigenetic and transcriptional mechanisms of ANCO1 as a tumor suppressor and highlighted its role as a chromatin regulator. It would be interesting to verify these findings in other breast cancer subtypes and disease stages and determine whether restoration of ANCO1 signaling could slow or prevent early-stage progression of breast cancer. 

## 4. Materials and Methods

### 4.1. Approval of Studies Involving Humans and Patient Informed Consent

The breast cancer samples used for TMA analysis were collected from invasive ductal breast cancer patients of three different subtypes (ER-positive, HER2-positive, and TNBC) who received surgeries at Medstar Georgetown University Hospital between 2004 and 2017. Clinical and demographical data were provided by the Histopathology and Tissue Shared Resource at Georgetown University Medical Center through the REDCap Database. The 150-patient representative group was selected from a database containing over 2000 research-consented breast cancer patients. The patients’ health information was de-identified to protect privacy. Protocols 1992-048, 2007-0345, and Pro00000007 were approved by the Georgetown University Medical Center Review Board for the construction of the microarray.

### 4.2. TMA Staining

TMA analysis was performed by the Histopathology and Tissue Shared Resource at the Georgetown University Medical Center. Breast tumor tissues obtained from breast cancer biopsies and reduction mammoplasty were embedded in paraffin. Two cylindrical cores (1.5 mm in diameter) per patient were embedded from morphologically representative regions of primary tumor blocks. Sections of 5 µm were prepared, and IHC staining was performed by the Georgetown University Histopathology and Tissue Shared Resource, utilizing standard procedures described elsewhere [62]. In brief, antigens were treated with citrate buffer at pH 6 (ThermoFisher Scientific, Waltham, MA, USA, 00-5000) and exposed to 3% hydrogen peroxide (Fisher Scientific, H325-500). To prevent non-specific binding of the antibody, the tissues were blocked with 10% normal goat serum. ANCO1 protein expression was detected by incubating the tissue sections overnight at 4 °C with a mouse monoclonal antibody at 1:100 dilution (Santa Cruz Biotechnology, Inc., Dallas, TX, USA, sc-81049). The primary antibody was detected using the DAKO Envision Plus HRP kit. The slides were then counterstained with hematoxylin solution (Sigma-Aldrich, Burlington, MA, USA, MHS16).

### 4.3. TMA Analysis Using Vectra3

Stained TMA slides were examined under the Vectra 3 Multi-Spectral Imaging Microscope. inForm software was used for ANCO1 level quantification. Cores of low quality were excluded from the analysis. The high ANCO1 threshold was determined using the following method: A core with high-intensity ANCO1 nuclei staining was selected as an example of positivity. The software then calculated an optical density score for each cell composing the core, which allowed for threshold parameters to be set by the software. A high ANCO1 numerical value threshold was set so that the majority of high-intensity ANCO1-positive cells were designated as ‘positive’. Cells with scores higher than this threshold were defined as ANCO1-high cells. The same strategy was used to determine a low ANCO1 threshold. A medium ANCO1 threshold was determined as the average of high and low thresholds. Percentages of cells of different ANCO1 levels were quantified in each core. The entire cohort comprised the average of the nuclear and cytoplasmic expression, and positive ANCO1 protein expression was summarized using median values (interquartile ranges (IQRs)). Survival analyses were performed using the ‘survminer’ R package (version 0.4.9) [63].

### 4.4. Gene Expression Analysis

The Molecular Taxonomy of Breast Cancer International Consortium (METABRIC) and Cancer Genome Atlas (TCGA) datasets [16,17] were used for the analysis of ANCO1 mRNA levels in different breast cancer subtypes. Gene expressions of 2509 breast tumor samples in the METABRIC dataset were analyzed by the Illumina Human v3 microarray, and 825 samples in the TCGA dataset were analyzed by the Agilent microarray. Totals of 1257 and 526 samples from Metabric and TCGA, respectively, with ANCO1 expression data and relevant clinical information, were included, and the analysis was performed using the “dplyr” R package (version 1.1.1) [64]. The IHC subtypes of breast tumors were determined by the HER2, ER, and PR statuses available in the dataset. 

### 4.5. Cell Culture

MCF10A (RRID:CVCL_0598) and MCFDCIS (RRID:CVCL_5552) cell lines [32] were maintained in DMEM/F12 (1:1) medium (Gibco, Waltham, MA, USA, 11039-021) with 5% horse serum, 20 µg/mL epidermal growth factor (EGF), 100 µg/mL hydrocortisone, 10 µg/mL insulin, and 100 ng/mL cholera toxin. HEK293T cells were maintained in DMEM medium (Gibco, 11995-065) with 10% fetal bovine serum (FBS). All cells were cultured at 37 °C with 5% CO_2_.

### 4.6. Cell Transfection and Infection

To make ANCO1 knockdown and control lentiviruses, HEK293T cells were transfected with plasmids containing target shANCO1 or control constructs together with pPACKF1 FIV packaging and pVSV-G envelop plasmids (See Table 1 for plasmid information). Cell transfection was performed using FuGENE6 (Promega, Madison, WI, USA, E2692) according to the manufacturer’s instructions. Briefly, the plasmids were mixed with FuGENE reagents and then applied to HEK293T cells for 4–6 h. Cell supernatants containing lentiviruses were collected at 48 and 72 h after transfection.

10A and DCIS cell lines in 6-well plates were infected with 1 mL of shANCO1 or control lentiviruses to generate shANCO1 and control lines in the presence of 4 µg/mL polybrene (Sigma-Aldrich). Infected cells were selected using 5 µg/mL puromycin for 4 days. Infected cells were then analyzed for ANCO1 levels and used for subsequent experiments. For coculture settings, shANCO1 DCIS cells were labeled with the pCDH-EF1-Luc2-P2A-tdTomato plasmid (RRID: Addgene_72486) by lentiviral infection.

### 4.7. Real-Time Quantitative PCR (RT-qPCR)

RNA was extracted from the cells with DNase digestion using the RNeasy Mini Kit (Qiagen, Hilden, Germany, 74106) according to the manufacturer’s instructions. RNA sample concentration and quality were measured using Nanodrop. Reverse transcription was conducted with the iScript cDNA Synthesis Kit (Biorad, Hercules, CA, USA, 1708891) according to the manufacturer’s instructions. RT-qPCR of gene expression was performed using the iQ SYBR Green Supermix (Biorad, 1708882) and realplex2 eppendorf PCR machine. Primers were obtained from Integrated DNA Technologies. (See Table 2 for primer sequences). Fold changes were calculated by subtracting ACTIN Ct values from ANCO1 Ct values and normalizing shANCO1 to control conditions (2^−ΔΔCT^).

### 4.8. Western Blotting

Cells were lysed using RIPA buffer consisting of 50 mM Tris-Cl (pH 8.0), 1% Triton X, 0.5% sodium deoxycholate, 0.1% SDS, and 150 mM NaCl for 15 min on ice and then centrifuged at 15,000× *g* for 10 min to remove debris. Protein concentration in the supernatants was quantified using the Bradford assay (Biorad, 5000006) with a spectrophotometer. Tris-acetate running buffer (Novex, Waltham, MA, USA, LA0041) and 3–8% tris-acetate NuPAGE gel (Invitrogen, Waltham, MA, USA, EA0375BOX) were used for electrophoresis, and proteins were transferred to PVDF membranes (Invitrogen, IB24002) with the iBlot 2 dry blotting system under 20 V for 10 min. The membranes were subsequently blocked with 5% milk and incubated with ANCO1 (Abcam, Cambridge, UK, ab50852, 1:50) or ACTIN (Millipore, Burlington, MA, USA, MAB1501, 1:15,000) antibodies overnight at 4 °C. Membranes were then washed and incubated with secondary antibody (GE Healthcare, Chicago, IL, USA, NA931V, 1:10,000) for 1 h at room temperature. The Immobilon Western Chemiluminescent HRP Substrate (Millipore, WBKLS0500) was used for signal development. Quantification was performed with ImageJ.

### 4.9. IF Staining

Cells were grown on glass coverslips for IF staining. Cells were fixed with 4% formaldehyde for 15 min and permeabilized with 0.1% Triton-X-100 for 10 min. Blocking was performed using 2% bovine serum albumin (BSA) with goat serum for 30 min. The above operations were carried out at room temperature. Cells were then incubated with ANCO1 antibody (Santa Cruz Biotechnology, Inc., Dallas, TX, USA, sc-81049, 1:50) in 2% BSA overnight at 4 °C on a shaker. Cells were then washed and incubated with Alexa Fluor 488-conjugated secondary antibody (Invitrogen, 1:200) and 4′,6-diamidino-2-phenylindole (DAPI) (Life Technologies, Carlsbad, CA, USA, S33025, 1:300) in 2% BSA for 1 h at room temperature. After washing, cells were mounted with Prolong Gold Antifade reagent (Invitrogen, P36930) on slides. Images were acquired with the Olympus IX-71 Inverted Epifluorescence Microscope. The fluorescence intensity was quantified with ImageJ.

### 4.10. FISH Analysis

The cells attached to slides were fixed twice in 3:1 methanol:glacial acetic acid and stored at −20 °C. The FISH protocol was derived from Pinkel and Gray [65]. The slides were washed twice in 2× Saline Sodium Citrate (SSC) and dehydrated with a graded ethanol series of 70%, 80%, and 95% for 2 min each at room temperature. The samples were then denatured using a 72 °C formamide solution (2× SSC and 70% formamide in water) in a glass Coplin jar for 2 min and dehydrated in a graded ethanol series as above. The XCE17 probe (MetaSystems, Boston, MA, USA) directly labeled with a Spectrum green dye was applied to the samples, coverslipped, and allowed to hybridize to the cells overnight at 37 °C in a humid chamber. The slides were post-washed for 2 min in 0.5× SSC at 72 °C and for 5 min in 2× SSC with 0.005% Tween at room temperature. The slides were counterstained with DAPI, cover-slipped, and examined with a Zeiss Axioskop fluorescence microscope equipped with Applied Imaging software. Images were taken using 10X ocular and 100X objective lenses under a confocal microscope.

### 4.11. SA β-Galactosidase Analysis

β-Galactosidase staining was conducted using the Senescence β-Galactosidase Staining Kit (Cell signaling, Danvers, MA, USA, 9860S) according to the manufacturer’s instructions. In brief, the cells were fixed and stained for β-Galactosidase as instructed and kept at 37 °C without CO_2_ overnight (10A cell lines) or for two days (DCIS cell lines) to develop blue coloration. Images were acquired with the Olympus IX-71 Inverted Epifluorescence Microscope, and quantification of positive cells was performed with ImageJ. 

### 4.12. Apoptotic Analysis

Cells were cultured in 10 cm dishes for 2 days until 70% confluency. Attached and floating cells were collected and analyzed by the Georgetown University Flow Cytometry and Cell Sorting Shared Resource. In brief, cell samples were incubated with 100 µL Annexin V Binding Buffer, 4 µL Annexin FITC (or AF647), and 4 µL PI (or Sytox blue) for 15 min at room temperature in the dark. Samples were then analyzed using a BD LSRFortessaTM Cell Analyzer.

### 4.13. Sphere Formation Assay 

The sphere formation assay was conducted as described previously [24]. Briefly, 8-well glass chamber CultureSlides (Falcon, Dublin, OH, USA, 354108) were coated with 100% Matrigel Basement Matrix, Reduced Growth Factor (Corning, Somerville, MA, USA, 354230) and allowed to solidify at 37 °C for 30 min. Cells were collected and resuspended in assay medium consisting of DMEM/12 (1:1), 2.5% horse serum, 100 µg/mL hydrocortisone, 10 µg/mL insulin, and 100 ng/mL cholera toxin. A total of 5000 cells with 2% Matrigel and 5 ng/mL EGF were seeded in each well on top of a solidified pure Matrigel layer. Cells were allowed to form spheres at 37 °C with 5% CO_2_. Images were taken with the Olympus IX-71 Inverted Epifluorescence Microscope. Images for sphere size quantification were taken on days 3 (10A) and 4 (DCIS). Quantification was performed with ImageJ (version 2.9.0/1.53t). 

### 4.14. Sphere Invasion Assay

Cell aggregates were first formed in U-shaped 96-well plates (Costar, New York, NY, USA, 7007) or 81-well agarose molds [66] and embedded in a mixture of 50% Matrigel and 50% Collagen I (3.38 mg/mL) in 8-well glass chamber CultureSlides to form spheres. A total of 1 ng/mL EGF was supplied. Images were acquired using the Olympus IX-71 Inverted Epifluorescence Microscope. Cells protruding from sphere boundaries and invading the surrounding ECM were defined as invading. Invasion area and traveling distance were quantified with ImageJ. For coculture settings, the labeled shANCO1 DCIS cells were mixed with unlabeled parental cells at a ratio of 1:4.

### 4.15. Animal Experiments 

The mice used in this study were maintained in the Georgetown University Division of Comparative Medicine, and compliance with the ethical standards approved by the Georgetown University Institutional Animal Care and Use Committee was ensured. Six-to-eight-week-old female athymic nude mice purchased from Envigo were injected subcutaneously with 750,000 DCIS control or shANCO1 cells. DCIS control or shANCO1 cells were resuspended in 1:1 PBS and Matrigel mix. The mice were euthanized 48 days after injection, and xenograft tumors were collected. Formalin-fixed and paraffin-embedded sections were stained with hematoxylin and eosin (H&E) or ANCO1 antibody (Santa Cruz, sc-81049) using the IHC methods described above.

### 4.16. RNA-seq

RNA was extracted from cells in 2D culture for 2 days or in 3D Matrigel for 5 days with DNase digestion using the RNeasy Mini Kit (Qiagen, 74106) according to the manufacturer’s instructions. RNA samples were quantified with Nanodrop, and the integrity of RNA was assessed with the Agilent 2100 Bioanalyzer. The library preparation and next-generation sequencing (NGS) were performed at Novogene Corporation Inc. (Sacramento, CA, USA). The 150bp paired-end sequencing was performed on a Novaseq sequencer with an average depth of 43 million paired reads per sample. 

Raw fastq files from RNA-seq experiments underwent quality checking with fastqc and were aligned to GRCh38 with hisat2 (version 2.1.0). The resulting BAM files were sorted with samtools (version 1.10-1-gc5b7134). Expression counts were calculated with featureCounts (version 1.6.3) and subsequently processed with the ‘edgeR’ package (version 3.40.2) in RStudio according to the developer’s manual [67,68,69]. Heatmaps were generated using the “Heatplus” package (version 3.6.0) [70], and volcano plots were generated using “dplyr” (version 1.1.1) [64], “ggplot2” (version 3.4.2) [71], and “ggrepel” (version 0.9.3) [72]. Processed expression values were analyzed for signaling pathway enrichment in GSEA (RRID:SCR_003199) or IPA (RRID:SCR_008653). RNA-seq tracks were visualized using the UCSC Genome Browser [73]. All RNA-seq experiments were performed in triplicate.

### 4.17. ChIP-seq

Cells were crosslinked with 1% Formaldehyde (Thermo Scientific, Waltham, MA, USA, 28908) for 15 min at room temperature. 2M glycine was added to stop crosslinking. Cells were collected in cold PBS with phosphatase and protease inhibitors (Roche, 04906837001, 04693159001). Lysed nuclei were sonicated using a Bioruptor Pico sonicator. An aliquot of fragmented chromatin was taken from each sample as input control. After pre-clearance, chromatin was incubated with 5 µL H3K27Ac antibody (Active Motif, Carlsbad, CA, USA, 39135) with shaking at 4 °C overnight. Samples were subsequently incubated with blocked Protein A/G beads (Thermo Scientific, 20423) with shaking at 4 °C for 4 h. Eluted chromatin and input samples were de-crosslinked in 200 mM NaCl and Protease K at 65 °C overnight. DNA was then purified using the PCR Purification Kit (Qiagen, 28106) according to the manufacturer’s instructions. The library preparation was performed using the NEBNext Ultra II DNA Library Prep Kit and Multiplex Oligos for Illumina (New England Biolabs, Ipswich, MA, USA, E7645S, E7335S). Library samples were sent to Novogene Corporation, Inc. for paired-end NGS using an Illumina Hiseq 4000 sequencer. 

The quality of raw fastq files was assessed with fastqc and aligned to GRCh38 with Burrows–Wheeler Aligner (BWA) (version 0.7.17) and Rsubread. The resulting BAM files were sorted and deduplicated with samtools. Differential genomic binding was established using the “csaw” package (version 1.32.0) in RStudio [74,75]. Gene annotation was performed with the “ChIPpeakAnno” package (3.0.0) [76,77]. Detection *q*-values were calculated by comparing ChIP-seq data to the input control using the macs3 (v3.0.0a6) bdgcmp function (-m qpois) [78]. This method uses the BH process for Poisson *p*-values to calculate the score in any bin using the control sample as lambda and the treatment (IP’d) sample as observation. Heatmaps and average plots were prepared using deepTools (v3.5.1) with the computeMatrix, plotHeatmap, and plotProfile functions [79]. We used default parameters with —referencePoint center and 3Kb margins. Overlaps of peaks were calculated using Bedtools (v2.26.0). Pathway enrichment analysis was performed using IPA or DAVID (RRID:SCR_001881). Motif enrichment analysis was performed using HOMER (v4.11.1) [80] and the findMotifsGenome.pl function. Sequencing data were converted to bigwig format using deepTools bamCoverage (—normalizeUsing RPKM—binSize 25—centerReads) [79]. ChIP-seq and RNA-seq data were uploaded as custom tracks for visualization on the UCSC genome browser [73]. All ChIP-seq experiments were performed in triplicate.

### 4.18. Statistical Analysis

Statistical analyses were performed with the R platform (version 4.2.1) using indicated packages from Bioconductor (RRID:SCR_006442) or Prism 7 (Graphpad Inc., San Diego, CA, USA, RRID:SCR_002798). Analysis of variance was used for multiple comparisons, and *t*-tests were used for unpaired comparisons. The chi-squared test was used for comparisons of frequency data. For the survival analyses of the TMA cohort, individual TMA cores were matched to the corresponding patient IDs. Patients’ overall survival and recurrence-free survival times were plotted by Kaplan–Meier estimation. Statistical tests for survival analyses based on ANCO1 mRNA expression were performed with the KM plotter (https://kmplot.com/analysis/, accessed on 23 December 2022). Statistical tests for enriched pathways were performed in GSEA, DAVID, or IPA, and those for motif analyses were performed in HOMER. *p*-value < 0.05 was used for statistical significance.

## Figures and Tables

**Figure 1 ijms-24-11505-f001:**
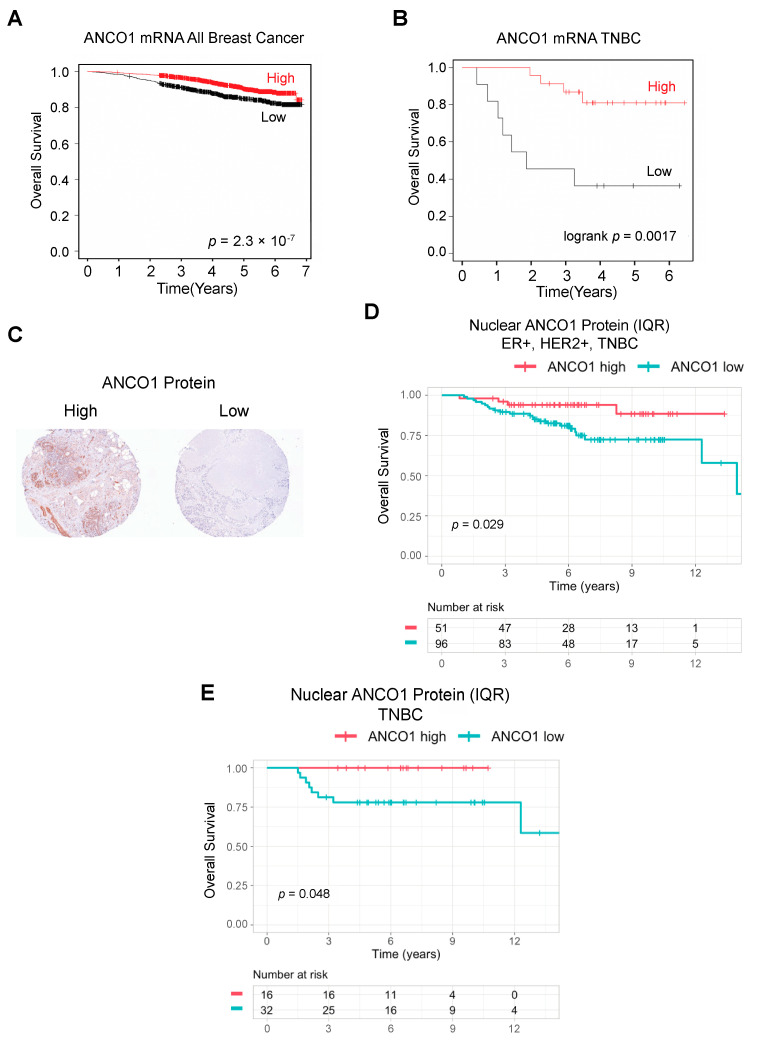
High ankyrin repeat domain containing 11 (ANCO1) expression is a positive prognostic marker in breast cancer patients. KM plots showing survival probability of patients with all subtypes of breast cancer (**A**) or triple-negative breast cancer (TNBC) without chemotherapy treatment (**B**) based on ANCO1 expression. Patients were stratified by ANCO1 mRNA levels. *n* = 2976 and 34, respectively. (**C**) Representative images of high (left) and low (right) ANCO1 protein expression in human breast tissue microarray (TMA). (**D**) Overall survival of patients with ER+, HER2+, and TNBC breast cancer based on nuclear ANCO1 protein levels. ANCO1 levels were stratified by interquartile ranges (IQRs). *n* = 147 patients. (**E**) Overall survival of patients with TNBC based on nuclear ANCO1 protein expression. ANCO1 levels were stratified by IQRs. *n* = 48 patients.

**Figure 2 ijms-24-11505-f002:**
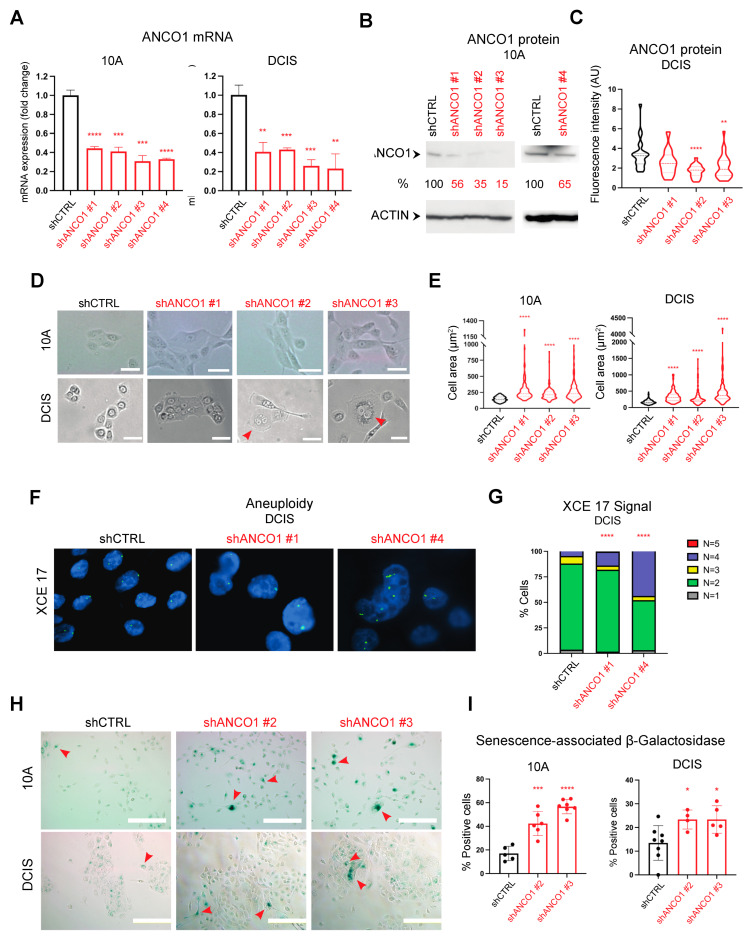
Reduction in ANCO1 levels causes aberrant cell morphology and leads to aneuploidy and senescence. (**A**) Verification of ANCO1 mRNA decrease in ANCO1 knockdown 10A and DCIS cell lines by RT-qPCR. Fold changes are relative to shCTRL groups. (**B**) Western blot of ANCO1 protein in 10A shANCO1 cell lines. Percentages indicate protein levels compared to control cells. The image for the ACTIN band was taken at 3 s exposure, and that for ANCO1 was taken at 1 min. (**C**) Quantification of ANCO1 immunofluorescence (IF) staining in shANCO1 and shCTRL DCIS cell lines. At least four fields of ~20 cells in total were quantified per group. AU = arbitrary units. (**D**) Bright-field images showing changes in the size and shape of shANCO1 10A and DCIS cells. Red arrowheads denote enlarged cells with more than one nucleus. Scale bars, 25 μm. (**E**) Quantification of cell size of 10A and DCIS shANCO1 and control cell lines. At least four fields of ~150 cells in total were quantified per group. (**F**) Fluorescence in situ hybridization (FISH) analysis for XCE17 shows aneuploid cells in shANCO1 DCIS cell lines. Images were taken under 10× ocular and 100× objective lenses. (**G**) Quantification of XCE17 FISH signals showing 4N cells in shANCO1 cell lines. A total of ~450 cells were quantified per group. (**H**) β-galactosidase staining of control and shANCO1 10A and DCIS cells. Red arrowheads denote examples of positive staining. Scale bars, 100 μm. (**I**) Quantification of β-galactosidase staining in 10A and DCIS shANCO1 lines compared to their controls. At least four fields of ~550 cells in total were counted per group. Error bars, means ± SDs. Unpaired *t*-tests were used for statistical analysis in panels (**A**,**C**,**E**,**I**). Fisher’s exact test was used in panel (**G**). * *p* < 0.05; ** *p* < 0.01; *** *p* < 0.001; **** *p* < 0.0001.

**Figure 3 ijms-24-11505-f003:**
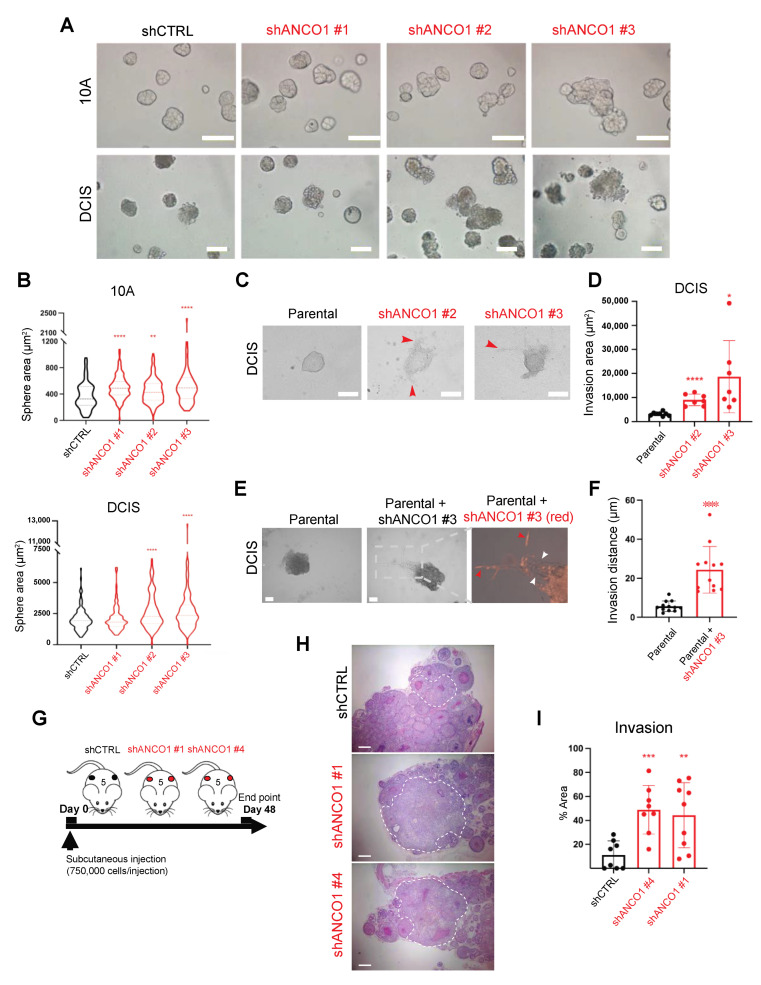
Reduction in ANCO1 levels causes aberrant sphere formation and enables invasion in vitro and progression in vivo. (**A**) Representative images of three-dimensional (3D) spheres formed by control or shANCO1 10A and DCIS cells (see Appendix A for schematic). Images were taken at day 3 (10A) and day 6 (DCIS). Scale bars, 50 μm. (**B**) Quantification of sphere size. A total of 83–196 spheres per group were quantified. Size quantification was based on 2D images. (**C**) Representative images of 3D spheres formed by parental or shANCO1 DCIS cells (see Appendix A for schematic). Cells were embedded in a mixture of Matrigel and Collagen I. Red arrowheads denote examples of invasive regions. Scale bars, 100 μm. (**D**) Quantification of invasion into the matrix around the spheres. Seven to eight spheres were quantified per group. (**E**) Images of 3D spheres generated from parental-only or mixed shANCO1 (red) and parental (unlabeled) DCIS cells at a 1:4 ratio. Spheres were embedded in a mixture of Matrigel and Collagen I. Red arrowheads denote examples of invading shANCO1 cells. White arrowheads denote examples of invading parental cells. Scale bars, 10 μm. (**F**) Quantification of traveling distances of parental DCIS cells from sphere borders in parental-only culture or mixed culture with shANCO1 cells. A total of 12 invasion arms were quantified in each group. (**G**) Experimental scheme of subcutaneous tumor growth from DCIS control or shANCO1 cell lines in athymic nude mice. Tumors were collected and analyzed at day 48 after injection. Each group consisted of five mice. (**H**) Representative images of hematoxylin and eosin (H&E) staining in mouse xenograft tumors. White circles indicate examples of invasive lesions. Scale bars, 500 μm. (**I**) Quantification of invasive lesions in each group. At least eight fields were quantified per group. Error bars, means ± SDs. Unpaired *t*-tests were used for statistical analysis in panels (**B**,**D**,**F**,**I**). * *p* < 0.05; ** *p* < 0.01; *** *p* < 0.001; **** *p* < 0.0001.

**Figure 4 ijms-24-11505-f004:**
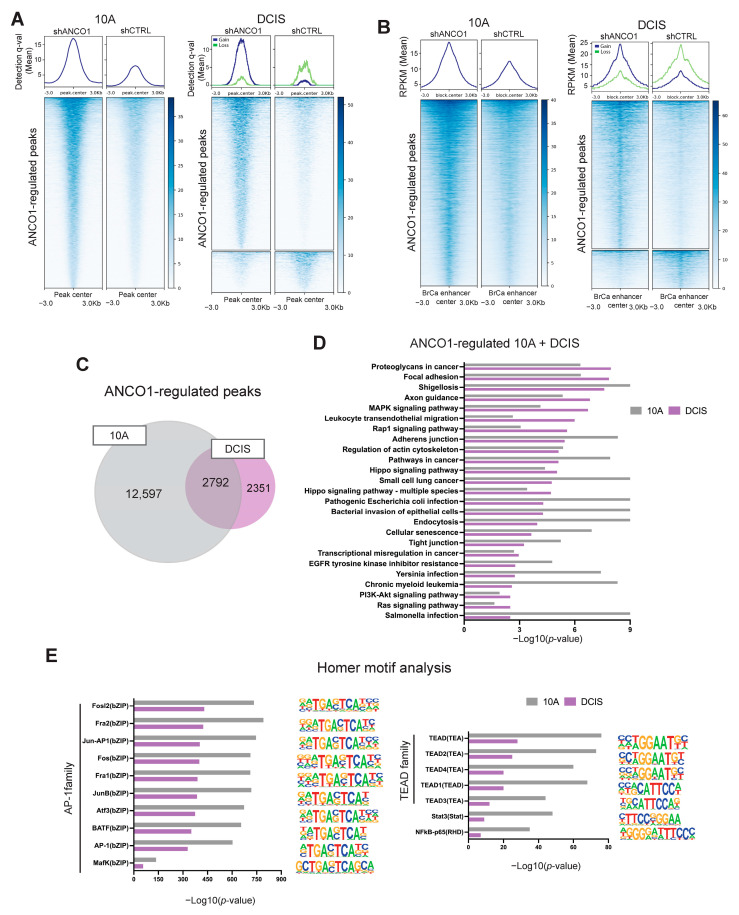
ANCO1 reduction increases chromatin accessibility and enhances H3K27Ac binding at known breast cancer enhancers. (**A**) Heatmaps and summary plots of detection *q*-values of ANCO1-regulated H3K27Ac peaks in 10A and DCIS shANCO1 versus control cell lines overlaid to the center of the peaks. Blue and green curves in the summary plot for the DCIS line indicate up- or down-regulated peaks in shANCO1 cells, respectively. The shANCO1 #3 cell line was used for this analysis. The 43 down-regulated peaks in 10A shANCO1 cells were not included in the heatmap due to the small number. (**B**) Heatmaps and summary plots of read counts of ANCO1-regulated H3K27Ac peaks in 10A and DCIS shANCO1 and control cell lines overlaid to known breast cancer enhancer regions [32]. (**C**) Venn diagram showing unique and shared ANCO1-regulated H3K27Ac peaks in 10A and DCIS cell lines. (**D**) DAVID pathway analysis results of top 25 enriched pathways in genes annotated to ANCO1-regulated H3K27Ac peaks in 10A and DCIS cell lines. Ranking was based on −log_10_ (*p*-value) values in the DCIS line. (**E**) Selected HOMER motif enrichment analysis results of up-regulated H3K27Ac peaks in shANCO1 versus control cells. The known motif method was used for discovery.

**Figure 5 ijms-24-11505-f005:**
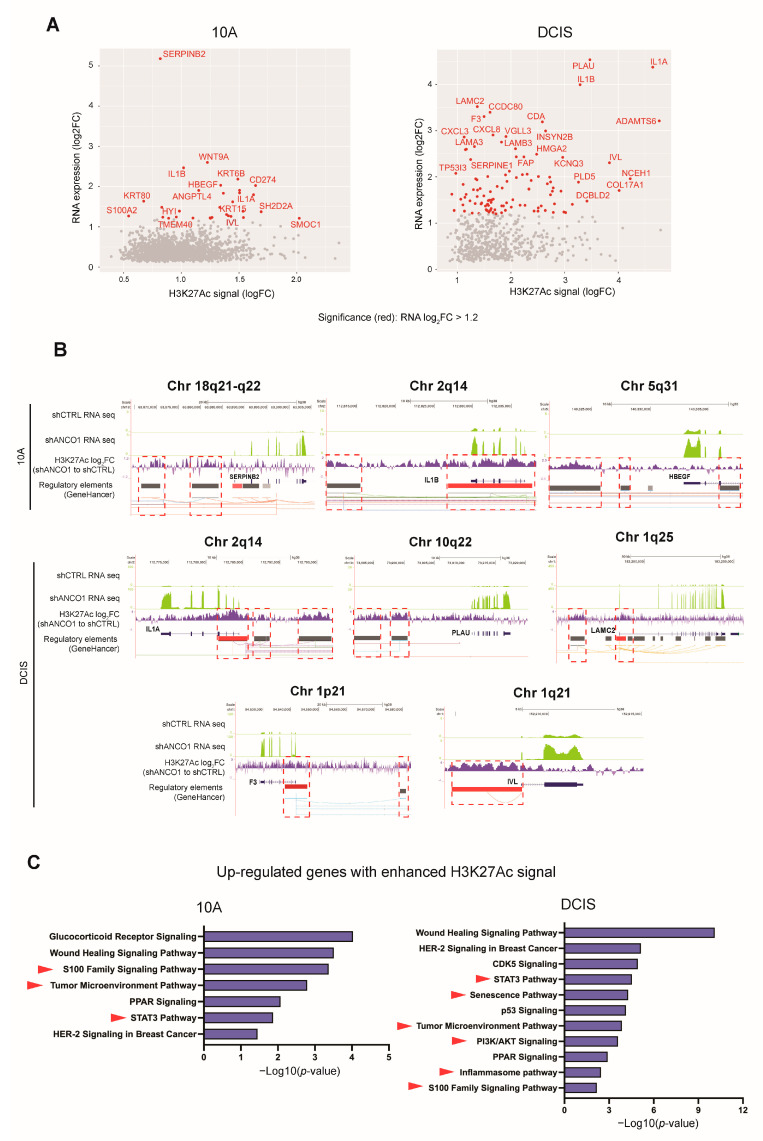
Loss of ANCO1 expression leads to increased H3K27Ac engagement at up-regulated gene sites. (**A**) Scatter plots of up-regulated genes with enhanced H3K27Ac signals in 10A and DCIS shANCO1 versus control cell lines. The shANCO1 #3 cell line was used for this analysis. (**B**) Examples of up-regulated genes in shANCO1 versus control cells with H3K27Ac ChIP-seq peaks at corresponding DNA regulatory elements. Green tracks are normalized RNA-seq reads from shANCO1 or control groups. Purple tracks show log_2_FCs (shANCO1 to control) of H3K27Ac peaks. Boxed areas show increased H3K27Ac signals at regulatory elements, with red coloring indicating promoters and grey coloring indicating enhancers. (**C**) Selected IPA results of up-regulated genes with increased H3K27Ac signals in 10A and DCIS ANCO1 knockdown cells.

**Table 1 ijms-24-11505-t001:** Plasmid information.

Plasmid	Manufacturer	Category Number
pLKO.1 ANCO1 shRNA #1	Sigma-Aldrich	TRCN0000140052
pLKO.1 ANCO1 shRNA #2	Sigma-Aldrich	TRCN0000144209
pLKO.1 ANCO1 shRNA #3	Sigma-Aldrich	TRCN0000140474
pLKO.1 ANCO1 shRNA #4	Sigma-Aldrich	TRCN0000344174
pLKO.1 GFP shRNA	A gift from David Sabatini (Addgene)	30323
pVSV-G	A gift from Robert Weinberg (Addgene)	8454
pPACKF1 FIV	System Biosciences	LV100A-1
pCDH-EF1-Luc2-P2A-tdTomato	Addgene	72486

**Table 2 ijms-24-11505-t002:** Primer sequences.

Plasmid	Forward	Reverse
*ANCO1*/*ANKRD11* (human)	TTGATGAGGACGACGAGCAG	TGACAGGATACGATGGGACG
*ACTIN* (human)	CCTGGCACCCAGCACAAT	GCCGATCCACACGGAGTACT

## Data Availability

The RNA-seq and ChIP-seq data generated in this study have been deposited with the Gene Expression Omnibus (GEO) under the SuperSeries record GSE230063. The RNA-seq data can also be accessed through the SubSeries record GSE230062. The ChIP-seq data can be accessed through the SubSeries record GSE230061.

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
