# Peer review of "Loss of ANCO1 Expression Regulates Chromatin Accessibility and Drives Progression of Early-Stage Triple-Negative Breast Cancer"

_ijms, 2023, doi:10.3390/ijms241411505_

Round 1
Reviewer 1 Report
Summary:
An overall good paper about the role of ANKRD11/ANCO1 as a potential prognostic marker for clinical outcomes in breast cancer and particularly in early-stage TNBC cells. The authors did a good job presenting their work and describing the results. They emphasize on the role of ANCO1 on H3K27 acetylation at the promoter of specific oncogenic target genes. The authors drew the right conclusions from the experiments without hyperbolizing the results. Just a few comments regarding some of the experiments.
Comments:
1- Figure 1. panels D and E are difficult to read, increase the typeface size.
2- Figure 5-figure C is quite impossible to read the IPA pathways, resize the figure or put some data in the supplemental.
3- I suggest to check the WB quantification in figure 1B, the presented actin looks quite different from the original WB submitted in supplementary materials.
4- There are a few spelling mistakes across the paper, quick review of grammar.
In general, I highly appreciate the quality of the manuscript, I recommend it for publication in the International Journal of Molecular Sciences with minor revisions.
Reviewer 2 Report
The authors identify the protein ANCO1 as a prognostic marker for breast cancer clinical outcome. Low nuclear expression of ANCO1 predicts lower overall survival of patients with TNBC. shRNA KO for ANCO1 lead to aneuploidy, senescence and invasive capacity of MCF10A and MCFDCIS cells. RNASeq analysis of 2D and 3D shRNA cell lines reveal expression of genes involved in invasion, EMT and senescence. ChIP-seq analysis shows increase in H3K27Ac binding regulating HIPPO, PI3-AKT, senescence signaling in shRNA cell lines.
It is a complete work characterizing ANCO1 function from prognosis indicator to transcriptional regulator. It includes a high amount of data combining patient gene expression analysis, in vivo tests, ChIP seq and RNA-Seq experiments on cell models.
Here are some remarks:
1- Fig 2A : Y axis legend has to be completed with “mRNA expression” (fold change)
2- Fig 2C : Y axis legend fluorescence intensity. Unit has to be mentioned
3- Fig 2E : could cell area be expressed in µm2
4- Fig 2G: no SD are presented.
5- Fig 2: Expression level for shANCO#4 is not presented. Analysis were done with shANCO#1 to #4. Each experiment presented was performed with different shANCO1. Why not the choice of a pair of shANCO1# based on the ANCO1 protein level expression to perform the different tests?
6- Legend of supplementary figures are missing
7- FigS2A To compare the expression level of ANCO1 protein between DCIS cells and MCF10A cells, images of immunofluorescence of both cell lines should be included with shCTRL and the different shRNA.
8-Lanes 173 to 176 this section could be part of the discussion
9-Lane 181: Is the induction of apoptosis is observed with the other shANCO1. Only shANCO1#3 is shown and data are not convincing since the apoptotic cell number is very weak. The figures should be improved. The Y and X-axis are not the same in the two cell lines figures.
10-Fig 3 A and B what is exactly measured to determine the sphere area? The sphere area should be expressed in µm2 .
11-Lanes 632-635: this part of methodology could be included in the part 4.6 Cell transfection and infection with the plasmid included in the table 1
12-Fig S5: how the authors justify the combination of three (2D) or 2 (3D) shRNA cell lines to compare gene expression? From my point of view, pattern of gene expression is different for several analysis. For MCF10A 2D shRNA#1 profile is very different from shRNA#2 and 3. For DCIS 2D, shRNA#3 is different from shRNA#1 and 2. For DCIS 3D, shRNA#3 is different from shRNA#2.
13-Lane 393 Discussion
14-In the discussion, cell models are associated to TNBC (triple negative breast cancer) cells. MCF10A cells are noncancerous breast-tissue-derived cells and can not be associated with TNBC.
Two cell models have been used representative of a pre-malignant stage (MCF10A) and early-stage carcinoma (DCIS). This difference and the associated data obtained are not discussed.
